# Allosteric Inhibitors of Zika Virus NS2B-NS3 Protease Targeting Protease in “Super-Open” Conformation

**DOI:** 10.3390/v15051106

**Published:** 2023-04-30

**Authors:** Ittipat Meewan, Sergey A. Shiryaev, Julius Kattoula, Chun-Teng Huang, Vivian Lin, Chiao-Han Chuang, Alexey V. Terskikh, Ruben Abagyan

**Affiliations:** 1Institute of Molecular Biosciences, Mahidol University, Nakhon Pathom 73170, Thailand; 2Skaggs School of Pharmacy and Pharmaceutical Sciences, University of California San Diego, La Jolla, CA 92093, USA; 3Sanford-Burnham-Prebys Medical Discovery Institute, La Jolla, CA 92037, USA

**Keywords:** Zika virus protease inhibitors, allosteric inhibitors, Zika virus NS2B-NS3 protease, super-open conformation

## Abstract

The Zika virus (ZIKV), a member of the Flaviviridae family, is considered a major health threat causing multiple cases of microcephaly in newborns and Guillain-Barré syndrome in adults. In this study, we targeted a transient, deep, and hydrophobic pocket of the “super-open” conformation of ZIKV NS2B-NS3 protease to overcome the limitations of the active site pocket. After virtual docking screening of approximately seven million compounds against the novel allosteric site, we selected the top six candidates and assessed them in enzymatic assays. Six candidates inhibited ZIKV NS2B-NS3 protease proteolytic activity at low micromolar concentrations. These six compounds, targeting the selected protease pocket conserved in ZIKV, serve as unique drug candidates and open new opportunities for possible treatment against several flavivirus infections.

## 1. Introduction

The Zika virus (ZIKV) is an emerging, global pathogen declared as a Public Health Emergency of International Concern by the World Health Organization. Approximately 84,000 cumulative cases have been reported globally since 2016 [1,2,3,4]. The virus is transmitted by the mosquito species *Aedes aegypti* and *Aedes albopictus*, as well as through human contact [1,5,6,7]. The main symptoms of a ZIKV infection, including fever, rash, and conjunctivitis, are relatively mild and frequently asymptomatic in adults. However, ZIKV has been associated with microcephaly in children born to mothers infected with ZIKV during pregnancy, as evidenced by the four-times increase in reported microcephaly cases from the end of January to mid-November of 2016 compared to the same period in 2015 [8,9,10,11]. Moreover, there is also evidence that ZIKV infection may be linked with Guillain-Barré syndrome and other neuroinflammatory disorders, which cause autoimmune degeneration of the myelin sheath in the peripheral neurons of adults since these conditions were observed to rise during the ZIKV epidemic in Colombia during 2015–2017 [12,13,14,15]. Currently, there are no effective drugs against ZIKV or any other flavivirus infection [16]. Furthermore, vaccines against several Flavivirus infections may be problematic due to antibody-dependent enhancement phenomena, with the notable exceptions of Japanese Encephalitis and Yellow Fever [17,18]. 

The two-part ZIKV protease is essential for viral replication, designating it as a potential target for treatment. ZIKV nonstructural NS3 protein is a multi-functional protein with protease and helicase activities. The C-terminal 440 residue region of NS3 encodes NS3 helicase (NS3hel), which participates in RNA replication and RNA capping [19,20,21,22,23]. A central cytosolic ~50 residue fragment of the transmembrane NS2B protein forms a functional NS2B-NS3pro complex with the N-terminal ~170 residues of NS3, responsible for the cleavage of ZIKV polyprotein (Figure 1). The cleavage results in activating capsid (C), pre-membrane (pr), membrane (M), and envelope (E) structural, and nonstructural (NS1, NS2A, NS2B, NS3, NS4A, NS4B, and NS5) proteins [24,25]. The NS2B cofactor is required for the protease activity of NS3pro. ZIKV NS2B-NS3pro possesses high proteolytic activity but becomes enzymatically inactive if the NS2B cofactor is deleted [25,26]. In addition to the viral polyprotein processing, the presence of multiple copies of active viral protease inside the host cell cytoplasm may lead to irreversible damage to numerous host cell proteins. Thus, NS2B-NS3pro is a promising drug target for the treatment of ZIKV infection [16,26,27]. Targeting the viral protease has been shown as a successful therapeutic strategy for ZIKV as well as the infections caused by other members of the Flaviviridae family, including Dengue, West Nile, and Hepatitis C viruses [28,29,30,31,32,33].

Designing or screening for effective competitive inhibitors for flavivirus proteases has been challenging since strong and specific binding to S1 and S2 active sub-pockets requires hydrophilic and electrostatic interactions, usually associated with low membrane permeabilities [29]. Therefore, in our research, we sought to identify a novel type of noncompetitive inhibitor against NS2B-NS3pro that prevent the formation of the active ZIKV NS2B-NS3pro complex. We chose to target a recently identified transient pocket present in the crystal structure of WNV NS2B-NS3pro, termed the “super-open” conformation [34]. The structural analysis of WNV NS2B-NS3pro reveals that there are two distinct conformations based on the placement of NS2B cofactor: the “closed conformation”, in which the NS2B wraps around NS3 and is the active conformation, and the “super-open” conformation, in which the NS2B chain turns and binds to a small area behind the active site, resulting in its inactivation (see Figure 2). Here, we employed a large-scale Molsoft-ICM docking screen of approximately seven million compounds to find effective non-covalent inhibitors for the allosteric pocket of ZIKV NS2B-NS3pro, with six selected top candidates evaluated experimentally, and they were confirmed to inhibit the enzymatic activity of the viral protease. 

## 2. Materials and Methods

### 2.1. Reagents

Routine laboratory reagents were purchased from Thermo Fisher Scientific, US. The tested compounds were purchased from Chembridge Corp., San Diego, CA, USA.

### 2.2. Virtual Ligand Screening of the Compound Library

The structure of NS2-NS3pro was obtained from the X-ray crystal structure (PDB ID 5TFN) [34]. The “super-open” conformation and the transient pocket exposed in that conformation were identified by comparing 5LC0 (closed) with five other structures (PDB ID: 5TFN, 5TFO, 6UM3, 5T1V, and 2GGV) [34,35,36,37]. The single chain construct in 5TFN contained both NS2 and NS3 domains, and the NS2 domain was marked to map possible interactions of the selected inhibitors with the NS3 chain. A docking screen was performed against approximately 7 million small molecules from the eMolecules catalog [38], commercially available compounds that have not been reported as ZIKV protease inhibitors and are predicted to have low toxicity using Molsoft ICM 3.9-1e software [39,40,41]. The scored binding poses of small molecules were predicted by the Biased-Probability Monte Carlo optimizer in Molsoft ICM software, and binding affinity of small molecules to the receptor was ranked based on force field-based docking score extended with additional free-energy contributions. All scoring functions and pharmacokinetic properties prediction were performed using Molsoft ICM-Pro v3.9 [39,40,41]. 

### 2.3. Cloning and Purification of ZIKV NS2B-NS3pro Construct

The recombinant construct expressing ZIKV NS2B-NS3 protease with a 6xHisTag on its N-terminus was used to transform competent *E. coli* BL21 (DE3) Codon Plus cells obtained from Stratagene. Transformed cells were grown at 30 °C in LB broth containing carbenicillin (0.1 mg/mL). Cultures were induced with 0.6 mM IPTG for 16 h at 18 °C. Cells were collected by centrifugation, re-suspended in Tris-HCl buffer, pH 8.0, containing 1 M NaCl, and disrupted by sonication (30 s pulse and 30 s intervals; 8 pulses) on ice. The pellet was removed by centrifugation (40,000× *g*; 30 min). The construct was then purified from the supernatant fraction on a Ni-NTA Sepharose, equilibrated with 20 mM Tris-HCl buffer, pH 8.0, supplemented with 1 M NaCl. After washing out the impurities using the same buffer supplemented with 35 mM imidazole, the bound material was eluted using a 35–500 mM gradient of imidazole. The fractions containing the recombinant protein were combined, dialyzed against 20 mM Tris-HCl, pH 8.0, containing 150 mM NaCl, and stored at −80 °C until use. The purity of the material was tested by SDS gel-electrophoresis (12% NuPAGE-MOPS, Invitrogen, Waltham, MA, USA), followed by Coomassie staining and by Western blotting with anti-HisTag antibodies. 

### 2.4. Protease Activity Assay with Fluorescent Peptide

The peptide cleavage activity assay with the purified ZIKV NS2B-NS3pro samples was performed in 0.2 mL 20 mM Tris-HCl buffer, pH 8.0, containing 20% glycerol and 0.005% Brij-35. The cleavage peptide pyroglutamic acid Pyr-Arg-Thr-Lys-Arg-7-amino-4-methylcoumarin (Pyr-RTKR-AMC) and enzyme concentrations were 20 μM and 10 nM, respectively [26]. Reaction velocity was monitored continuously at λex = 360 nm and λex = 465 nm on a Tecan fluorescence spectrophotometer (Tecan Group Ltd., Männedorf, Switzerland). All assays were performed in triplicate wells of a 96-well plate. Dose–response curves and IC_50_ values of each compound were calculated accordingly using SciPy and Matplotlib Python packages.

### 2.5. Determination of the IC_50_ Values of the Inhibitory Compounds

The ZIKV NS2B-NS3pro construct (20 nM) was pre-incubated for 30 min at 20 °C with increasing concentrations of the individual compounds in 0.1 mL 20 mM Tris-HCl buffer, pH 8.0, containing 20% glycerol and 0.005% Brij 35. The Pyr-RTKR-AMC substrate (20 μM) was then added in 0.1 mL of the same buffer. All assays were performed in triplicate wells of a 96-well plate. IC_50_ values were calculated by determining the concentrations of the compounds needed to inhibit 50% of the NS2B-NS3pro activity against Pyr-RTKR-AMC. GraphPad Prism was used as fitting software.

## 3. Results

### 3.1. Identification of an Allosteric Site on ZIKV NS2B-NS3 Protease

The active site of ZIKV NS2B-NS3 protease has been a favorable protein target for designing small molecules against ZIKV infection. However, a recent analysis of the NS2B-NS3 protease structure suggested that the catalytic triad of ZIKV NS2B-NS3pro is conserved in various human serine proteinases, and the corresponding pocket is shallow. Thus, designing specific and potent competing small molecules against the active pocket of ZIKV NS2B-NS3pro is quite challenging. To identify a new promising druggable pocket, we analyzed four structures of ZIKV NS2B-NS3 protease that were recently deposited in the Protein Data Bank (PDB: 5TFN, 5TFO, 6UM3, and 5T1V) [25]. The protease in all four structures contained the novel “super-open” pocket. The active “closed” conformation observed in PDB ID 5LC0 [36] undergoes a transition to the inactive super-open conformation and reveals a targetable pocket. Therefore, we considered that interference between the NS3 domain and its cofactor, NS2B (Figure 2A), might be a superior drug target to stabilize the inactive “super-open” conformation in the design of allosteric novel highly potent and specific small molecules for ZIKV infection treatment. The list of contact residues in the newly identified transient pocket includes W83, L85, V146, A87, A88, I147, G148, D86, L149, V154, V155, I123, D120, T118, G153, N152, Q74, D75, and L76. The transient pocket was then used as the receptor to find strong binders using the virtual docking screen.

### 3.2. Virtual Docking Screen of Seven Million Compounds Targeting the Allosteric Hidden Site of ZIKV NS2B-NS3 Protease

The resulting model of the “super-open” state of ZIKV protease (PDB 5TFN) was docked and scored against approximately seven million small molecules from the eMolecules database. Chemicals were first filtered by removing compounds with high toxicity propensity based on over 1000 structural alerts [41]. The binding free energy of each compound to the allosteric pocket was estimated by a docking score computed via the MolSoft ICM-Pro package [39,40]. The ICM scoring function includes van der Waals potential for a hydrogen atom probe; van der Waals potential for a heavy-atom probe (generic carbon of 1.7 Å radius); optimized electrostatic term; hydrophobic terms; and lone-pair-based potential for approximation of the intermolecular interaction between the receptor and ligand. Ten different compounds with the top ICM-Docking scores were initially tested in the cell-based assay to assess their inhibition activity against ZIKV NS2B-NS3 protease (see Appendix A). 

### 3.3. Selection of Core Functional Group

Ten protease inhibitor candidates suggested from predicted docking poses and binding scores were purchased from an available vendor (ChemBridge Corp, 11199 Sorrento Valley Rd., Suite 206, San Diego, CA, USA) and tested against 10 nM of ZIKV NS2B-NS3 in proteolytic activity assays with 20 μM of the fluorogenic peptide substrate, Pyr-RTKR-AMC, in order to validate viral protease inhibition. The structures and docking scores of the initial ten protease inhibitor candidates found from the virtual screening can be found in Appendix A. We found that compounds with phenylquinoline and aminobenzamide groups, shown in Figure 3, demonstrated desirable activity, indicating a potential scaffold for further optimization of allosteric inhibitors of ZIKV NS2B-NS3 protease. The binding conformation of RI07, the representative of phenylquinoline and aminobenzamide substituent compounds, in the deep allosteric pocket of the “super-open” conformation of NS2B-NS3 is shown in Figure 2B,C, suggesting strong binding affinity between aminobenzamide and hydrophobic residues including W83, L85, D86, A87, A88, V146, and L149, in the identified allosteric pocket of ZIKV NS2B-NS3.

### 3.4. Variation of Phenylquinoline and Aminobenzamide Substituents and Enzymatic Activities against ZIKV NS2B-NS3 Protease

Phenylquinoline and aminobenzamide-containing compounds showed high affinity to ZIKV NS2B-NS3 in proteolytic activity assays. This prompted us to search for more inhibitors from the same chemical class that are commercially available. We found five compounds from the available vendor with structures related to RI07, including the presence of aminobenzamide and phenylquinoline with varied substituents. These compounds, labeled as RI22, RI23, RI24, RI27, and RI28, were tested for their inhibition activity in the enzymatic assay. The structures of RI07 and its derivatives can be found in Figure 3.

The novel compounds in this series were evaluated for their ability to inhibit the proteolytic activity of the ZIKV protease. The inhibitor candidates, with IC_50_ values ranging in low micromolar concentrations from 3.8 to 14.4 µM, were identified (see Table 1 and Figure 4). Table 1 lists the docking scores, molecular weight, and IC_50_ values of all candidates as well as their relevant pharmacokinetic and toxicity properties, including water solubility, cell permeability, propensity for pan-assay interference, potassium channel blocking activity, polar surface area, and toxicity. These properties were evaluated using appropriate Molscreen models from the MolSoft ICM-Pro package [39,40,41]. The predicted parameters for these potential candidates fell within acceptable ranges. According to the docking simulation, RI07, the best compound in this series, was predicted to interact with several residues in the novel “super-open” pocket of the ZIKV NS2B-NS3 protease, including W83, L85, V146, A87, A88, I147, G148, D86, L149, V154, V155, I123, D120, T118, G153, N152, Q74, D75, and L76. The docking poses of all six compounds in this series, found in Appendix A, demonstrate consistent binding conformations in the identified allosteric pocket. The results suggest that targeting the allosteric pocket, which is necessary for ZIKV NS2B-NS3 complex formation, with compounds containing phenylquinoline and aminobenzamide groups has a high potential for further research on the development of anti-flaviviral agents (Table 2).

## 4. Discussion

ZIKV NS2B-NS3 protease is a two-component chymotrypsin-like serine protease consisting of the NS2B cofactor and the NS3 protease domain, similar to other Flaviviridae members. The active site of ZIKV NS2B-NS3pro includes three conserved amino acid residues forming a classic catalytic triad (His51, Asp75, and Ser135 in ZIKV) [18]. Due to its importance in the virus life cycle and propagation, NS2B-NS3pro is a promising target for antiviral drug design. Unfortunately, because of its shallow and solvent-exposed pocket and high structural homology among the active centers of multiple cellular serine proteases and the viral NS2B-NS3pro, the development of inhibitors with high cellular permeability, stability, and selectivity targeting the active site is challenging [44]. Thus, through the design of inhibitors targeting new allosteric sites on ZIKV NS2B-NS3pro, we can bypass this obstacle and develop effective therapeutic approaches.

The analysis of recently deposited protein structures found two conformations of ZIKV NS2B-NS3pro, which are the inactive “super-open” conformation and the active “closed” conformation. All structural rearrangements of ZIKV protease in “super-open” conformation are incompatible with the protease’s catalytic activity. However, this catalytic activity of NS2B-NS3pro can be completely restored by transitioning back to the “closed” conformation. Transitioning from ZIKV NS2B-NS3pro closed conformation to super-open conformation reveals a transient pocket at the interface between NS2B and NS3 in the closed conformation. Our purpose was to show that the major reorganization of the NS3pro C-terminal loop creates a transient novel druggable pocket that can be used for the development of specific binding, disrupting the protease activity allosterically (Figure 2A). This transient pocket may be present in proteases of other flaviviruses (e.g., PDB ID 2GGV for the West Nile virus protease) [34]. These novel structures demonstrate the detachment of the C-terminal part of NS2B from a deep pocket. The contact residues in the targeted pocket have favorable characteristics, such as shape, depth, and hydrophobicity, for drug-like small molecules. 

The virtual screening results showed that compounds containing phenylquinoline and aminobenzamide were predicted to have negative binding free energy to the “super-open” pocket on ZIKV NS3pro, indicating thermodynamically favorable ligand–protein complex formation. The binding pose of RI07, the top compound in this series according to the docking study and its IC_50_ value, suggests that the 2-aminobenzamide group in the prospective molecules is particularly effective at interacting with the side chains of L76, W83, L85, V146, I147, G148, and L149 in the hydrophobic region. Its two amide oxygens also form hydrogen bonds with amine hydrogens of V155 and N152 at distances of 3.2 and 2.6 Å, respectively. The docking pose of other compounds, compared to RI07, shows similar binding poses (Appendix A) and comparable hydrogen bonding distances (Appendix A), demonstrating the consistency of binding conformations in the identified allosteric pocket. On the other hand, the substituted phenylquinoline functional group, compared to the aminobenzamide group, interacts with a smaller number of residues, such as Q74, T118, D120, and I123, but its rigidity and unique structure fit exceptionally well in this narrow part of the pocket as shown in Figure 2B. 

The estimated binding free energies of all six compounds in this series from the docking study correspond to the evaluated IC_50_ values ranging from 3.8 to 17.4 μM, as shown in Table 1, making them suitable inhibitor candidates against the novel allosteric pocket. It is evident that selected derivatives of RI07 did not improve the IC_50_ compared to RI07, and the IC_50_ values vary significantly with different substituent groups on the phenylquinoline functional group; this underscores the potential for further optimization to increase activity, especially with this functional group. In addition to the enzymatic activity assay, the predicted key chemical properties, such as water solubility, cell permeability, propensity for pan-assay interference, potassium channel blocking activity, polar surface area, and ICM ToxScore, are all within acceptable ranges for being drug-like molecules.

These findings confirmed our hypothesis that this novel allosteric pocket revealed transiently in the “super-open” conformation can be used as a target for a new type of antiviral for the Zika virus. In addition to inhibiting ZIKV protease, the identified compounds containing phenylquinoline and aminobenzamide in this series may be applied against other flaviviruses such as WNV, DENV, TBEV, and YFV by exploiting the presence of “super-open” pockets in their protease domains. This target is beneficial because of its conservation, hence, its lower propensity for treatment escaping mutations, and its favorable shape. The identified allosteric inhibitors targeting ZIKV NS2B-NS3 protease and preventing viral proliferation may be further optimized for improved efficacy, pharmacokinetics (PK), pharmacodynamics (PD), and reduced adverse side-effect profile.

## 5. Conclusions

Infections with ZIKV are current global health concerns due to the risk of microcephaly in newborns and Guillain-Barré syndrome in adults. Since there is no approved anti-ZIKV agent and vaccination strategies are limited due to antibody-dependent enhancement (ADE) effects observed in Dengue and other flaviviruses, The discovery of small-molecule orally available drugs as ZIKV infection treatments has been limited. We have reported a new strategy of targeting a “super-open” conformation of ZIKV NS2B-NS3 protease suggested by the analysis of existing X-ray crystal structures of WNV and ZIKV proteases. The newly identified pocket is a preferable target compared to an active site due to its shape, conservation, and hydrophobicity. We have identified six novel compounds containing phenylquinoline and aminobenzamide that inhibited ZIKV NS2B-NS3 protease at IC_50_ values ranging from 3.8 to 14.4 μM. The predicted pharmacokinetic properties of all compounds were also evaluated, showing promising drug-like properties. Targeting the identified transient pocket may prove to be a promising strategy for fighting other flaviviral infections as well due to the similar pattern of inactive-to-active state transitions of the viral NS2B-NS3 system.

## Figures and Tables

**Figure 1 viruses-15-01106-f001:**
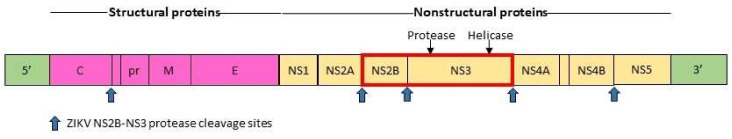
Schematic representation of the ZIKV genome. ZIKV polyprotein precursor includes structural proteins, shown in magenta, and nonstructural proteins, shown in yellow. ZIKV NS2B-NS3pro, highlighted by a red box, plays a significant role in viral protein activation. Its cleavage sites are represented by blue arrows.

**Figure 2 viruses-15-01106-f002:**
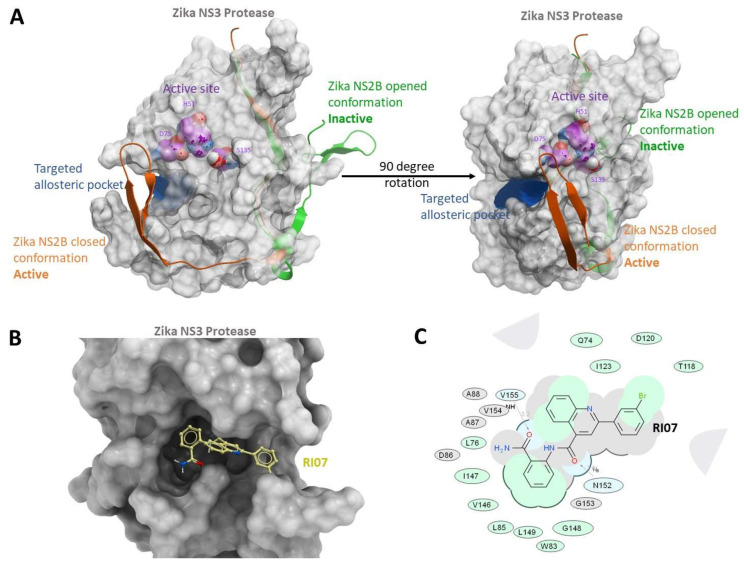
Targeting a hidden pocket exposed in the “super-open” conformation of ZIKV NS2B-NS3 protease. (**A**) ZIKV NS3 subunit in grey and active site in purple. Beta hairpin of NS2B shown in closed conformation opens a transient (allosteric) pocket by moving away into an open (green ribbon) conformation. Targeted allosteric pocket is in blue, and catalytic triad of NS3, S135, H51, and D75 are labeled and shown in CPK; (**B**) predicted 3D binding pose of RI07 targeting the super-open conformation of NS2B-NS3 protease; and (**C**) 2D interaction diagram for predicted pose of RI07 compound (Q74, D75, L76, W83, L85, D86, A87, A88, T118, D120, I123, V146, I147, G148, L149, N152, G153, V154, and V155). Hydrogen bonding interactions are shown by grey dotted lines.

**Figure 3 viruses-15-01106-f003:**
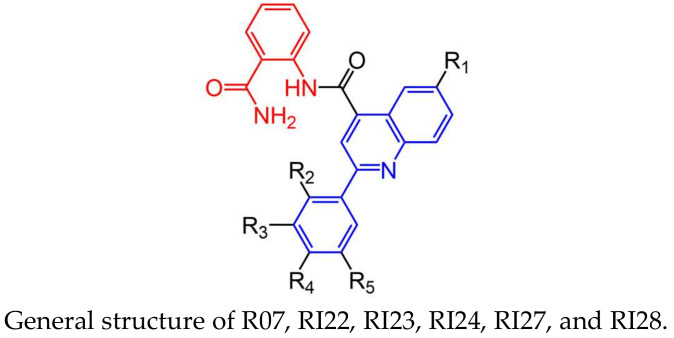
The figure presents the chemical structures of a general scaffold and six specific phenylquinoline and aminobenzamide compounds investigated in this study. Phenylquinoline substructures are highlighted in blue, and aminobenzamide substructures are highlighted in red.

**Figure 4 viruses-15-01106-f004:**
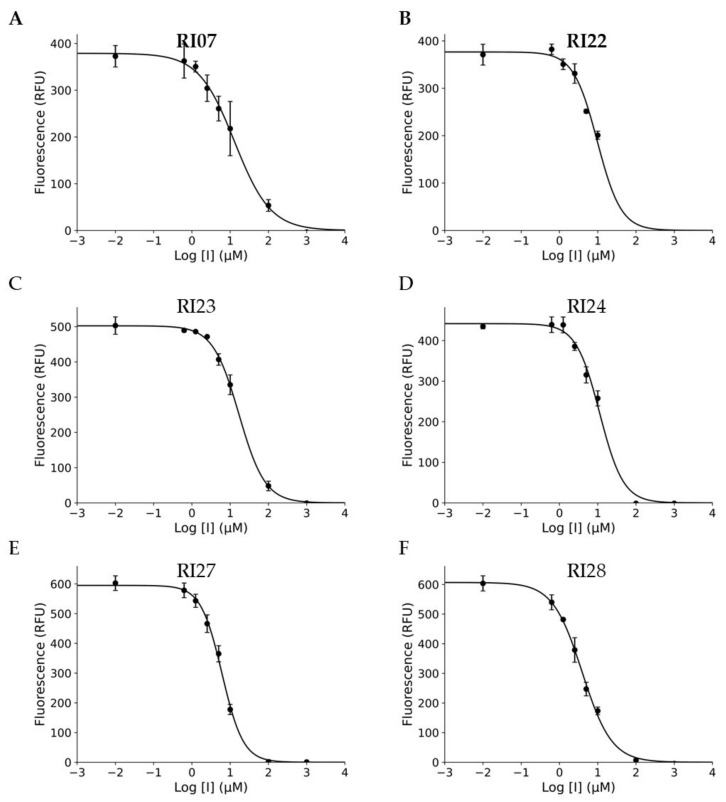
Six experimentally validated inhibitors of catalytic activity of Zika virus protease and their dose–response curves. The dose–response inhibition of NS2B-NS3 protease by RI07, RI22, RI23, RI24, RI27, and RI28 and the IC_50_ values of the compounds against ZIKV NS2B-NS3 protease were measured in protease cleavage assay with Pyr-Arg-Thr-Lys-Arg-MCA fluorescent peptidic substrate (Peptides International-MRP-3159-v) [26]. SciPy and Matplotlib Python packages were used for curve fitting (**A**–**F**). The *X*-axis shows decimal logarithms of compound concentrations in micromolar units versus the peptidic substrate cleavage in RFUs.

**Table 1 viruses-15-01106-t001:** IC_50_ values, binding score against ZIKV NS2B-NS3 protease, and pharmacokinetic parameters for compounds in this study.

Compounds ID	IC_50_ (μM)	Molecular Weight (g/mol)	ICM Binding Score ^a^	Predicted Pharmacological Properties ^b^								
				Solubility molLogS ^c^	Lipophobicity molLogP ^d^	Druglikeness ^e^	molCACO2 ^f^	molPAMPA ^g^	molHERG ^h^	molPAINS ^i^	Polar Surface Area (molPSA) ^j^	Tox_Score ^k^
**RI07**	3.8	445	−37	−5.12	5.07	0.57	−5.09	−4.71	0.13	0.05	115	0
**RI22**	6.2	381	−31	−4.50	4.59	0.59	−4.99	−4.71	0.11	0.01	110	0
**RI23**	11.9	435	−31	−5.92	5.28	0.70	−5.08	−4.80	0.37	0.06	108	0.42
**RI24**	17.4	445	−30	−5.54	5.17	0.93	−5.07	−4.70	0.09	0.04	116	0
**RI27**	10.5	415	−33	−4.65	5.16	0.78	−5.10	−4.63	0.09	0.04	104	0.42
**RI28**	14.4	411	−29	−4.45	4.14	0.40	−5.26	−4.61	0.07	0.02	156	0

^a^ Binding score was calculated using the Dockscan function in ICM-Pro [39,40]. The binding scores are calculated based on the binding free energy between the receptor and ligand. Lower scores suggest strong complex binding. ^b^ The prediction of relevant pharmacological properties was calculated using the Chemical Properties prediction function in ICM-Pro [41]. ^c^ Water solubility (molLogS) in 10-based logarithms of the solubility in M. ^d^ Lipophilicity (LogP) is the logarithm of the ratio of the compound concentration in octanol and water. ^e^ Druglikeness, value ranging between −1 and 1. A higher number indicates more drug-like properties. ^f^ CACO-2 permeability; value over −5 indicates high permeability. ^g^ PAMPA permeability, value over −5 indicates high permeability. ^h^ hERG inhibition; value over 0.5 indicates high probability of being an hERG inhibitor that may block potassium ion channels in the heart. ^i^ Pan Assay Interference Compound (PAINS), value over 0.5 indicates high probability of being a PAIN compound. ^j^ Polar surface area in square angstroms. ^k^ Toxicity scores calculated based on the presence of known toxic substructures within the compounds, value over 1 indicates that the molecule contains unfavorable substructures or substituents, suggesting a potentially toxic compound.

**Table 2 viruses-15-01106-t002:** IC_50_ values of the compounds reported to target the active site of ZIKV NS2B-NS3 protease and the compounds tested against the “super-open” pocket of ZIKV NS2B-NS3 protease in this study.

Compounds ID	IC_50_	Comments	Targets	Ref
**Compound 1**	0.2	Dipeptide inhibitor	**Active site**	[25,42]
**Compound 3**	4.1	Nonpeptidic small molecule inhibitor contains sulfonamide and benzothiazole groups	**Active site**	[25,42]
**Bromocriptine**	21.6	Repurposing dopamine receptor agonist as a ZIKV protease inhibitor	**Active site**	[43]
**RI07**	3.8	445	“Super-open” pocket	This study
**RI22**	6.2	381	“Super-open” pocket	This study
**RI23**	11.9	435	“Super-open” pocket	This study
**RI24**	17.4	445	“Super-open” pocket	This study
**RI27**	10.5	415	“Super-open” pocket	This study
**RI28**	14.4	411	“Super-open” pocket	This study

## Data Availability

All data generated or analyzed during this study are included in this published article.

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
