# Peer review of "Allosteric Inhibitors of Zika Virus NS2B-NS3 Protease Targeting Protease in “Super-Open” Conformation"

_viruses, 2023, doi:10.3390/v15051106_

Round 1

Reviewer 1 Report

The manuscript is focused on the search for potential allosteric inhibitors against the Zika Virus. The target of the study is the NS2B-NS3 protease in a “Super-Open” conformation. The screened compounds followed well-designed strategies that allowed identifying important structures/pharmacophores that have potential against the virus, as the tests resulted in the inhibition of the target protein activity with low IC50.

The manuscript in general is very well written, and its methodology is well elaborated and sufficient to validate its hypotheses. The results are well presented, as well as the discussion with very relevant counterpoints to emphasize its importance.

Therefore, the work as submitted is suitable for publication in this journal.

Author Response

Reviewer #1

The manuscript is focused on the search for potential allosteric inhibitors against the Zika Virus. The target of the study is the NS2B-NS3 protease in a “Super-Open” conformation. The screened compounds followed well-designed strategies that allowed identifying important structures/pharmacophores that have potential against the virus, as the tests resulted in the inhibition of the target protein activity with low IC50.

The manuscript in general is very well written, and its methodology is well elaborated and sufficient to validate its hypotheses. The results are well presented, as well as the discussion with very relevant counterpoints to emphasize its importance.

Therefore, the work as submitted is suitable for publication in this journal.

Response: We thank Reviewer #1 for the time, effort, and positive evaluation of our manuscript. We hope that our research will contribute to the development of new therapies against Zika Virus infections.

Reviewer 2 Report

The study by Meewan and colleagues used in silico docking of a compound library to identify possible inhibitors targeting an allosteric site in the ZIKV NS2B-NS3 protease. They then tested derivatives based on their strongest candidates  for protease inhibition in vitro and ZIKV replication inhibition in cell culture and could identify a series of compound acting antiviral that target an allosteric site in the protease and possibly stabilize the inactive „super-open“ conformation of the enzyme.

This is an interesting study and highly relevant topic since there is currently no available drug to treat ZIKV or other flavivirus infections.  While this reviewer cannot judge the quality of the structural interpretation and of the molecular docking data, there are however important flaws in the antiviral assay depicted in Figure 5 that need to be addressed. Furthermore, the authors miss control inhibitors to control their assays and put their results into perspective. Last, the manuscript needs to be improved to be more didactic, in particular for the public of virologists (the main audience of Viruses), and to correct a number of language mistakes.

Major comments:

1.      The authors propose that the allosteric site targeted might be more promising than the active site, but experiments are missing to test this assumption. In particular, the authors should test known ZIKV protease inhibitors – in particular inhibitors targeting the active site - in parallel to their candidates. At least one positive control should be used for the protease and antiviral assays. Regarding Table 1, it would be useful here to incorporate a broader range of known inhibitors, including the one(s) used as controls in the experimental part.

2.      The antiviral assay presented in Figure 5 is not convincing.

The authors should show a representative picture (Dapi, ZIKV channel, merge) for each condition.

Please explain what the individual dots represent (pictures?) and how many individual experiments and pictures per experiments were analyzed. Please indicate the unit in Fig. 5B.

It would be more appropriate to use the Dapi signal to count the number of cells in the different conditions (segmenting their nuclei) rather than simply looking at the total Dapi signal, which could vary due to difference in staining efficiency between samples.

The authors should also use an alternative and complementary cytotoxicity assay since dead / unhealthy but adherent cells would be counted with the presented approach. There are many commercially available kits to do so.

In the text, the authors seem to gloss over the cytotoxicity effects observed, although they are substancial for some compounds (even with the Dapi readout used which is likely underestimating the toxicity). Please describe the cytotoxic effects.

Based on cytotoxicity and antiviral activity at 10 uM, the authors should select 1-2 candidates to test more precisely in a dose response and describe IC50, IC90, EC50, EC90 and therapeutic index.

3.      The authors do not provide enough details explaining how they ended up with the selected 7 molecules. They should also adapt to the public of mostly virologists reading Viruses by better explaining the structural information and docking data.

Section 3.2, Line 178: Please explain what the 10 classes of compounds are.

Section 3.3: It is not clear here what the 10 tested compounds are. Please provide the compound information and a summary of the inhibition data. Was RI07 one of this compound? Could these compounds be included in Table 1?

Section 3.4: The authors derive new inhibitors from RI07, one of the most promising compound from the 10 initial compound tested, is this correct? The authors should write that the chemical derivation did not improve the anti-protease activity.

Please include the color code and symbol legend in the 2D interaction diagrams and also, what are the residues listed in the legend.

Figure 3: here it would be useful to highlight the phenylquinoline and aminobenzamide groups and substitutions and give more explanations in the legend.

Table 1: Some of the scores are well explained (e.g. Drug likeness), with indications how to interpretate them, but these indications are missing for others (e.g. is a good ICM binding score low or high?). Some units are missing (e.g. IC50, MW). Color coding of the scores (e.g. blue for best scores, red for worst scores) or at least highlighting the scores over the relevant threshold (e.g. in bold characters) might help the reader interpretate this table.

Is potassium channel blocking or possibility to be an hERG inhibitor desirable or not?

4.      Section 2.4 and Figure 4: Please cite a reference for the protease assay with the Pyr-RTKR-AMC peptide if this assay is accepted and already described. If this is not the case, the authors should provide more data to validate the assay, including an experiment with a control peptide that is not cleaved by the protease and/or with a protease-null mutant, but also including a positive control inhibitor, as mentioned above already.

5.      The authors try to expand their results to other flaviviruses  (e.g. in the abstract, they indicate that the 6 compounds target thte ZIKV protease and other flaviviruses) but this is not supported by any data. Please remove this over-interpretation and rather discuss this point in the conclusion.

Minor comments:

- The authors should discuss in more details the „super-open“ conformation of the protease, in particular whether this conformation was already accepted, whether their data support this model, whether the allosteric site was already targeted in inhibitor screens, but also how the protease is believed to shift between these conformations.

- Please briefly justify the use of NPCs for the antiviral assay.

- Please indicate in the Methods section how the ZIKV stock was obtained or amplified.

- The virus inhibition assay is not accurately described: The assay used does not quantify the number of ZIKV copies“ (line 248) or the „number of virus / amount of virus“ (Lines 322-323, line 324), it simply evaluates the immunostaining signal of a particular viral protein.

- Line 329: The authors suggest that modifications in RI22 and RI27 might affect interaction in the protein pocket. Should this be visible in Figure S2? If yes, can the authors highlight this? Also, why not showing the interaction diagram for RI27?

- Lines 337-345: The authors should elaborate and explain more precisely their arguments to propose that other flaviviruses also have a super open pocket in their protease domain (with reference to the literature, or data on aminoacid conservation, etc). Line 342-343: do the authors mean conservation among flaviviruses or among different strains of ZIKV? Please provide numbers to help the reader judge about this conservation. For instance, how is the described binding pocket conserved among Flaviviruses / ZIKV strains as compared to the protease active site?

- Line 354: The authors mention that they inspected the proteases of all flaviviruses, but according to line 79, only inspected ZIKV protease structures and 1 WNV protease structure. Also, how exactly was the WNV protease structure used in this manuscript?

- Figures S1: contact residues are partly difficult to read

- Figures S2-S6: distances are difficult to read

- Please verify the syllabification used in the text, e.g. activ-ity (line 48).

- The text needs a major re-reading since a number of mistakes have slipped. Some are listed below:

Line 35: no italics

Line 85: was ranked

Line 90: was used

Line 113: 20°C

Line 125: the cells were approximately 90% confluent

Line 139: an allosteric site

Line 154: in the newly

Line 156: transient pocket.... find strong

Line 164:  catalytic

Line 190: compounds used in

Lines 213-214: with several

Lines 214-215: repetition of previous pagraph (list of residues)? What is different in the contact residue prediction?

Line 229: between -1 and 1. Higher

Table 1: ICM Binding Score

Lines 235-236: unclear, please rephrase

Line 245: in RFUs

Line 248: using immunofluorescence to visualize ZIKV replication

Line 251: was plotted

Figure 5A, Y axis: Ratio of ZIKV protein /Dapi

Lines 264-265: Double word with “NPCs cells”, cells has to be removed

Line 292: reveals

Line 276: consisting of the NS2B

Lines 305-306: suggests that the.... in the prospective molecules...

Line 313: interacts

Line 339: Zika virus / ZIKV

Line 245: please define PK/PD

Lines 349-351: incomplete sentence

Line 353 quotation marks for super-open missing

Line 355: due to

Figure 3: Point at sentence end is missing

Author Response

Reviewer #2

The study by Meewan and colleagues used in silico docking of a compound library to identify possible inhibitors targeting an allosteric site in the ZIKV NS2B-NS3 protease. They then tested derivatives based on their strongest candidates for protease inhibition in vitro and ZIKV replication inhibition in cell culture and could identify a series of compound acting antiviral that target an allosteric site in the protease and possibly stabilize the inactive „super-open“ conformation of the enzyme.

This is an interesting study and highly relevant topic since there is currently no available drug to treat ZIKV or other flavivirus infections. While this reviewer cannot judge the quality of the structural interpretation and of the molecular docking data, there are however important flaws in the antiviral assay depicted in Figure 5 that need to be addressed. Furthermore, the authors miss control inhibitors to control their assays and put their results into perspective. Last, the manuscript needs to be improved to be more didactic, in particular for the public of virologists (the main audience of Viruses), and to correct a number of language mistakes.

Comment 1. The authors propose that the allosteric site targeted might be more promising than the active site, but experiments are missing to test this assumption. In particular, the authors should test known ZIKV protease inhibitors – in particular inhibitors targeting the active site - in parallel to their candidates. At least one positive control should be used for the protease and antiviral assays. Regarding Table 1, it would be useful here to incorporate a broader range of known inhibitors, including the one(s) used as controls in the experimental part.

Response: We thank Reviewer #2 for the valuable comments and suggestions. We agree that having an orthosteric protease inhibitor as a positive control would be nice to compare the efficacy of targeting the allosteric pocket in comparison with the traditional active site inhibitors. Unfortunately, we did not have access to the known ZIKV protease inhibitors. Instead, for comparison, we selected a few non-covalent inhibitors reported to target  the active site of ZIKV (listed in table 2) to show the range of IC50 values from those compounds. Those values were compared with IC50s of the identified inhibitors in this study in similar conditions. Additionally, we have provided results of a separate viral proliferation assay for chloroquine, which is reported to be active against ZIKV. The results for the antiviral assay shown in figure 5 have been modified and improved as suggested. 

Comment 2:

The antiviral assay presented in Figure 5 is not convincing.

The authors should show a representative picture (Dapi, ZIKV channel, merge) for each condition.

Please explain what the individual dots represent (pictures?) and how many individual experiments and pictures per experiments were analyzed. Please indicate the unit in Fig. 5B.

It would be more appropriate to use the Dapi signal to count the number of cells in the different conditions (segmenting their nuclei) rather than simply looking at the total Dapi signal, which could vary due to difference in staining efficiency between samples.

The authors should also use an alternative and complementary cytotoxicity assay since dead / unhealthy but adherent cells would be counted with the presented approach. There are many commercially available kits to do so.

In the text, the authors seem to gloss over the cytotoxicity effects observed, although they are substancial for some compounds (even with the Dapi readout used which is likely underestimating the toxicity). Please describe the cytotoxic effects.

Based on cytotoxicity and antiviral activity at 10 uM, the authors should select 1-2 candidates to test more precisely in a dose response and describe IC50, IC90, EC50, EC90 and therapeutic index.

Response: We appreciate this insightful feedback. We have decided to remove Figure 5 from this revised version since the experiment is incompleted.

Comment 3.  The authors do not provide enough details explaining how they ended up with the selected 7 molecules. They should also adapt to the public of mostly virologists reading Viruses by better explaining the structural information and docking data.

Section 3.2, Line 178: Please explain what the 10 classes of compounds are.

Section 3.3: It is not clear here what the 10 tested compounds are. Please provide the compound information and a summary of the inhibition data. Was RI07 one of this compound? Could these compounds be included in Table 1?

Section 3.4: The authors derive new inhibitors from RI07, one of the most promising compound from the 10 initial compound tested, is this correct? The authors should write that the chemical derivation did not improve the anti-protease activity.

Please include the color code and symbol legend in the 2D interaction diagrams and also, what are the residues listed in the legend.

Figure 3: here it would be useful to highlight the phenylquinoline and aminobenzamide groups and substitutions and give more explanations in the legend.

Table 1: Some of the scores are well explained (e.g. Drug likeness), with indications how to interpretate them, but these indications are missing for others (e.g. is a good ICM binding score low or high?). Some units are missing (e.g. IC50, MW). Color coding of the scores (e.g. blue for best scores, red for worst scores) or at least highlighting the scores over the relevant threshold (e.g. in bold characters) might help the reader interpretate this table.

Is potassium channel blocking or possibility to be an hERG inhibitor desirable or not?

Response: Thank you for pointing out the missing information about the initial ten candidates. We selected the initial ten candidates from the virtual in silico screening and ended up choosing one candidate, which is RI07, to pursue further due to its effective activity. We agree that the structural and inhibition data of those ten initial candidates should be provided too. Therefore, we have added the relevant data to the Supplementary Table S1. The appropriated changes have been also made in the revised manuscript on page 5 line 176 to 178: “Ten different compounds with the top ICM-Docking scores were initially tested in the cell-based assay to assess their inhibition activity against ZIKB NS2B-NS3 protease”. The most active compound RI07 based on the inhibition of ZIKV protease enzymatic activity was from the initial ten compounds. RI07 derivatives we tried did not improve the IC50 in this enzymatic activity assay. 

The discussions of that matter were added to the revised manuscript on page 10 line 304-307: 

“It is evident that selected derivatives of RI07 did not improve the IC50 from RI07 and the IC50 values vary significantly with different substituent groups on the phenylquinoline functional group; this underscores the potential for further optimization to increase activity especially with this functional group.”

Changes of Figure 3 were made according to the comment. We colored phenylquinoline and aminobenzamide substructures, in red and blue, respectively.

The corrections of the units, additional description of ICM Binding Scores, and hERG inhibition predictions have been added to Table 2 to address Reviewer #2 suggestion.

Comment 4.  Section 2.4 and Figure 4: Please cite a reference for the protease assay with the Pyr-RTKR-AMC peptide if this assay is accepted and already described. If this is not the case, the authors should provide more data to validate the assay, including an experiment with a control peptide that is not cleaved by the protease and/or with a protease-null mutant, but also including a positive control inhibitor, as mentioned above already.

Response: Thank you for a good comment. The procedure was adapted from the previously published experiment [1].  We have added the reference describing the protease assay in the section 2.4 and figure 4 in the revised version of the manuscript.

[1] S. A. Shiryaev et al., “Characterization of the Zika virus two-component NS2B-NS3 protease and structure-assisted identification of allosteric small-molecule antagonists.,” Antiviral Res, vol. 143, pp. 218–229, Jul. 2017, doi: 10.1016/j.antiviral.2017.04.015.

Comment 5.  The authors try to expand their results to other flaviviruses  (e.g. in the abstract, they indicate that the 6 compounds target thte ZIKV protease and other flaviviruses) but this is not supported by any data. Please remove this over-interpretation and rather discuss this point in the conclusion.

Response: Thank you for the comment. We have removed the mentioned interpretation from the abstract of the revised manuscript.

Minor comments:

Comment 6. The authors should discuss in more details the „super-open“ conformation of the protease, in particular whether this conformation was already accepted, whether their data support this model, whether the allosteric site was already targeted in inhibitor screens, but also how the protease is believed to shift between these conformations.

Response:  We appreciate this suggestion. In response, We have added sentences to explain the “super-open” conformation on page 2 lines 67 to 71: “The structural analysis of WNV NS2B-NS3pro reveals that there are two distinct conformations based on the placement of NS2B cofactor: the “closed conformation”, in which NS2B wraps around NS3 and is the active conformation, and the “super-open” conformation, in which the NS2B chain turns and binds to a small area behind the active site, resulting in its inactivation (see Figure 2).”

Comment 7. Please briefly justify the use of NPCs for the antiviral assay.

Response: This is a fair suggestion. In response, we have added a sentence to justify the use of human NPCs for the antiviral assay on page 11 lines 274 to 276: “Human NPCs are selected as host cells since they are the preferred target for ZIKV during infection, which can lead to severe neurological defects, including microcephaly.”

Comment 8. Please indicate in the Methods section how the ZIKV stock was obtained or amplified.

Comment 9.  The virus inhibition assay is not accurately described: The assay used does not quantify the number of ZIKV copies“ (line 248) or the „number of virus / amount of virus“ (Lines 322-323, line 324), it simply evaluates the immunostaining signal of a particular viral protein.

Comment 10.  Line 329: The authors suggest that modifications in RI22 and RI27 might affect interaction in the protein pocket. Should this be visible in Figure S2? If yes, can the authors highlight this? Also, why not showing the interaction diagram for RI27?

Comment 11.  Lines 337-345: The authors should elaborate and explain more precisely their arguments to propose that other flaviviruses also have a super open pocket in their protease domain (with reference to the literature, or data on aminoacid conservation, etc). Line 342-343: do the authors mean conservation among flaviviruses or among different strains of ZIKV? Please provide numbers to help the reader judge about this conservation. For instance, how is the described binding pocket conserved among Flaviviruses / ZIKV strains as compared to the protease active site?

Comment 12.  Line 354: The authors mention that they inspected the proteases of all flaviviruses, but according to line 79, only inspected ZIKV protease structures and 1 WNV protease structure. Also, how exactly was the WNV protease structure used in this manuscript?

Comment 13.  Figures S1: contact residues are partly difficult to read

Comment 14.  Figures S2-S6: distances are difficult to read

Comment 15.  Please verify the syllabification used in the text, e.g. activ-ity (line 48).

The text needs a major re-reading since a number of mistakes have slipped. 

Some are listed below:

Comment 16.  Line 35: no italics

Comment 17.  Line 85: was ranked

Comment 17.  Line 90: was used

Comment 18.  Line 113: 20°C

Comment 19.  Line 125: the cells were approximately 90% confluent

Comment 20.  Line 139: an allosteric site

Comment 21.  Line 154: in the newly

Comment 22.  Line 156: transient pocket.... find strong

Comment 23.  Line 164:  catalytic

Comment 24.  Line 190: compounds used in

Comment 25.  Lines 213-214: with several

Comment 26.  Lines 214-215: repetition of previous pagraph (list of residues)? What is different in the contact residue prediction?

Comment 27.  Line 229: between -1 and 1. Higher

Comment 28.  Table 1: ICM Binding Score

Comment 29.  Lines 235-236: unclear, please rephrase

Comment 30.  Line 245: in RFUs

Comment 31.  Line 248: using immunofluorescence to visualize ZIKV replication

Comment 32.  Line 251: was plotted

Comment 33.  Figure 5A, Y axis: Ratio of ZIKV protein /Dapi

Comment 34.  Lines 264-265: Double word with “NPCs cells”, cells has to be removed

Comment 35.  Line 292: reveals

Comment 36.  Line 276: consisting of the NS2B

Comment 37.  Lines 305-306: suggests that the.... in the prospective molecules...

Comment 38.  Line 313: interacts

Comment 39.  Line 339: Zika virus / ZIKV

Comment 40.  Line 245: please define PK/PD

Comment 41.  Lines 349-351: incomplete sentence

Comment 42.  Line 353 quotation marks for super-open missing

Comment 43.  Line 355: due to…

Comment 44.  Figure 3: Point at sentence end is missing

Response: All these comments were addressed. Thank you.

Reviewer 3 Report

The reviewer likes this approach and its potential as a study. This study forms the new basis of drug discovery as it relies upon binding of the drug molecule to allosteric site rather than the active site. Designing and developing antiviral drugs is a challenge for our scientific community. The study seems to be very interesting as it is the need of the era. The authors have successfully carried out virtual screening (molecular docking) and selected seven top hits and assessed their potential as antiviral agents through enzymatic and cell-based assays. Finally six antiviral compounds have been reported. However, a few major concerns need to be addressed before considering this manuscript for final publication.

I have listed these major issues below:

  1. In Introduction section, I will suggest to add the statistical epidemiological data of this viral disease as the statistics are changing at a good pace. Authors can refer to WHO dashboard and CDC.

  2. In section 2.2. authors highlight the selection of 7 million small molecules from eMolecules catalog. Authors are requested to detail out more about the type of molecules selected i.e. are these molecules FDA approved, whether they are undergoing clinical trials, reported small molecules. Throw some light on the database selected. Is this database freely available ? Is it similar to ZINC database?

  3. In section 3.2. authors mention “Ten different classes of compounds with the top ICM-Docking scores were initially tested in ZIKV NS2B-NS3 protease enzymatic assay described in Methods”. The data relating to docking must be presented in a tabular form with the top hits.

  4. In section 3.3. authors mention “Ten protease inhibitor candidates suggested from predicted docking poses and binding scores ......”. Kindly mention the docking related results in tabulated form along with major interactions observed. Are these the same molecules described in Section 3.2. line 178-180 ?

  5. Taking into consideration comments 3 and 4; are the terms classes and candidates used interchangeably in text ?

  6. In section 3.3. line 186-188: The structure of the potential scaffold should be presented separately, only then further optimization of allosteric inhibitors of ZIKV NS2B-NS3 protease can be justified.

  7. Authors are encouraged to provide the detailed description of all the methodologies employed in performing molecular docking. It will help enable the readers gain a deeper insight on how to carry out the study.

  8. In section 3.4. authors mention “That prompted us to search for more potent inhibitors from the same chemical class”. Did the authors perform molecular docking again to select the best hits. This needs a proper explanation.

  9. In section 3.4. authors mention “An extended set of compounds of various substituents on phenylquinoline was identified in a chemical vendor catalog to test in the same inhibition assay”. What does chemical vendor catalog signify? What does extended set of compounds signify ?

  10. Did the authors face any difficulties in solubilizing these compounds while carrying out antiviral assays ? Were there any precipitations observed ? as these compounds are from synthetic constructs.

  11. The presentation of the methods and results pertaining to performing molecular docking is not in ordered manner. Authors are requested to arrange it properly for a better understanding.

  12. In the provided supplementary file; The 2D interaction image of RI27 is missing and an addition image of R126 has been put up. Kindly check and put it up accordingly. I suppose the image presented is of R127 as per structure.

  13. Figure 4 : The compound code in figure legend (RI07, RI22, RI23, RI24, RI27, and RI28) does not match the compound code in the figure (RI07, RI22, RI23, RI24, RI26, and RI27). Kindly rectify the compound codes.

  14. The authors should thoroughly check the manuscript for  grammatical errors

Author Response

Reviewer #3 

The reviewer likes this approach and its potential as a study. This study forms the new basis of drug discovery as it relies upon binding of the drug molecule to allosteric site rather than the active site. Designing and developing antiviral drugs is a challenge for our scientific community. The study seems to be very interesting as it is the need of the era. The authors have successfully carried out virtual screening (molecular docking) and selected seven top hits and assessed their potential as antiviral agents through enzymatic and cell-based assays. Finally six antiviral compounds have been reported. However, a few major concerns need to be addressed before considering this manuscript for final publication.

I have listed these major issues below:

Comment 1. In Introduction section, I will suggest to add the statistical epidemiological data of this viral disease as the statistics are changing at a good pace. Authors can refer to WHO dashboard and CDC.

Response: We thank reviewer 3 for their comment. In response, we have added the epidemiological data of the viral diseases in the Introduction section on page 1 lines 33 to 38: “However, ZIKV has been associated with microcephaly in children born to mothers infected with ZIKV during pregnancy as evidenced by four times increase in reported microcephaly cases from the end of January to mid-November of 2016 compared to the same period in 2015 [8]–[11] Moreover, there is also evidence that ZIKV infection may be linked with Guillain-Barré syndrome and other neuroinflammatory disorders which causes autoimmune degeneration of the myelin sheath in the peripheral neurons of adults since these conditions were observed to rise during ZIKV epidemic in Colombia during 2015-2017 [12]–[15].”

Comment 2.  In section 2.2. authors highlight the selection of 7 million small molecules from eMolecules catalog. Authors are requested to detail out more about the type of molecules selected i.e. are these molecules FDA approved, whether they are undergoing clinical trials, reported small molecules. Throw some light on the database selected. Is this database freely available ? Is it similar to ZINC database?

Response: We appreciate your comment, and have added the description about the selected 7 million compounds from eMolecules catalog in page 3 lines 81 - 84: “A docking screen was performed against approximately 7 million small molecules from eMolecules catalog[36], commercially-available compounds that haven’t been reported as ZIKV proteases inhibitors and predicted to have low toxicity using Molsoft ICM software [37]–[39].”

Comment 3.  In section 3.2. authors mention “Ten different classes of compounds with the top ICM-Docking scores were initially tested in ZIKV NS2B-NS3 protease enzymatic assay described in Methods”. The data relating to docking must be presented in a tabular form with the top hits.

Response: We are grateful to Reviewer #3 for pointing this out. We added the structures, binding scores, and the initial evaluations of their antiviral activities and toxicity to Supplementary Table S1. The revised manuscript was modified on page 5, lines 182 to 186: “Ten different compounds with the top ICM-Docking scores were initially tested in the cell-based assay to assess their inhibition activity against ZIKB NS2B-NS3 protease (see Table S1).”

Comment 4.  In section 3.3. authors mention “Ten protease inhibitor candidates suggested from predicted docking poses and binding scores ......”. Kindly mention the docking related results in tabulated form along with major interactions observed. Are these the same molecules described

Response: We provided the structures and binding scores in the Supplementary Table S1. In addition, we made appropriate changes on page 6, lines 184-186: “The structures and docking score of initial ten protease inhibitor candidates found from the virtual screening can be found in Table S1.”

Comment 5.  in Section 3.2. line 178-180 ?

Taking into consideration comments 3 and 4; are the terms classes and candidates used interchangeably in text ?

Response: We thank Reviewer #3 for the comment. Originally, we used the term 'classes' to refer to each individual compound among the ten initial compounds identified in virtual screening, as they represent groups containing structurally related compounds. Now, to avoid confusion, we removed this term (‘classes’) from the revised manuscript as can be seen in page 5, line 176 to 178: “Ten different compounds with the top ICM-Docking scores were initially tested … “.

Comment 6.  In section 3.3. line 186-188: The structure of the potential scaffold should be presented separately, only then further optimization of allosteric inhibitors of ZIKV NS2B-NS3 protease can be justified.

Response: Thank you for the suggestions, we added potential scaffolds of phenylquinoline and aminobenzamide groups in Figure 3.

Comment 7.  Authors are encouraged to provide the detailed description of all the methodologies employed in performing molecular docking. It will help enable the readers gain a deeper insight on how to carry out the study.

Response: We agree with this suggestion.  The description of the molecular docking simulations was extended in page 5 lines 172 to 176: “The ICM scoring function includes, van der Waals potential for a hydrogen atom probe; van der Waals potential for a heavy-atom probe (generic carbon of 1.7A radius); optimized electrostatic term; hydrophobic terms; and, loan-pair-based potential for approximation of the intermolecular interaction between the receptor and ligand.”

Comment 8.  In section 3.4. authors mention “That prompted us to search for more potent inhibitors from the same chemical class”. Did the authors perform molecular docking again to select the best hits. This needs a proper explanation.

Response: Indeed, that was an unclear statement. We searched for available compounds similar to RI07. The changes have been made on page 7 lines 208 to 203: “That prompted us to search for more inhibitors from the same chemical class which are commercially available. “

Comment 9.  In section 3.4. authors mention “An extended set of compounds of various substituents on phenylquinoline was identified in a chemical vendor catalog to test in the same inhibition assay”. What does chemical vendor catalog signify? What does extended set of compounds signify ?

Response: To clarify the selection of derivatives of RI07 in this section, we have revised the paragraph on page 7 lines 203 to 208: “We have found that there are five compounds from the available vendor that have structure related to RI07 including the presence of aminobenzamide and phenylquinoline with varied substituents. Those compounds, labeled asRI22, RI23, RI24, RI27, and RI28, are tested for their inhibition activity in the enzymatic assay The structures of RI07 and its derivatives can be found in Figure 3.”

Comment 10.  Did the authors face any difficulties in solubilizing these compounds while carrying out antiviral assays ? Were there any precipitations observed ? as these compounds are from synthetic constructs.

Response: This is an important question. Initially, we had difficulties preparing these compounds at very high concentrations. Finally, we found a way of solving that problem by preparing a 100 mM stock compound in DMSO which is sufficient to generate a dose-response curve in an enzymatic assay. 

Comment 11.  The presentation of the methods and results pertaining to performing molecular docking is not in an ordered manner. Authors are requested to arrange it properly for a better understanding.

Response: Thank you for your suggestion. We have edited the results involving the molecular docking according to the reviewer's comment.

Comment 12.  In the provided supplementary file; The 2D interaction image of RI27 is missing and an addition image of R126 has been put up. Kindly check and put it up accordingly. I suppose the image presented is of R127 as per structure.

Response: We thank Reviewer #3 for pointing out these typos. In response, we corrected the compound code in the legend of Figure S5.

Comment 13.  Figure 4 : The compound code in figure legend (RI07, RI22, RI23, RI24, RI27, and RI28) does not match the compound code in the figure (RI07, RI22, RI23, RI24, RI26, and RI27). Kindly rectify the compound codes.

Response: We thank Reviewer #3 for pointing this out. In response, we have fixed an error on the figure legend.

Comment 14.  The authors should thoroughly check the manuscript for  grammatical error

Response: Thank you for your suggestion, we have improved the grammar in the revised manuscript.

Round 2

Reviewer 2 Report

The authors have addressed my comments with elaborated answers and changes in the manuscript text and figures. Although it is unfortunate that there is no antiviral assay in this new version (Figure 5 deleted), I hope that follow up studies will test the antiviral potential of these compounds. Please make sure to read another time the manuscript as some typos and errors remain or have slipped in the new sections, for instance, please rephrase the legend of Figure 3 (typos and grammatics) and correct the typo in the Legend of Table 2 („to target the active site“).

Author Response

Response to reviewer 2 Comments

Reviewer #2
The authors have addressed my comments with elaborated answers and changes in the manuscript text and figures. Although it is unfortunate that there is no antiviral assay in this new version (Figure 5 deleted), I hope that follow up studies will test the antiviral potential of these compounds. Please make sure to read another time the manuscript as some typos and errors remain or have slipped in the new sections, for instance, please rephrase the legend of Figure 3 asite“).

Response: We appreciate the feedback and suggestions from reviewer 2. We agree that the antiviral assay and the toxicity assessments are essential for new compounds as new antiviral treatment agents. We will definitely conduct follow-up studies of the compounds identified in this research.
In response to your comments, we have incorporated changes in the revised version of the manuscript. We have corrected the legend of Figure 3, Table 1, and Table 2. In addition, we have addressed the grammatical errors and typos throughout the revised manuscript. We hope that you will find this version suitable for publication in Viruses journal.

Sincerely,
Ruben Abagyan
April 26, 2023

Reviewer 3 Report

The manuscript is now acceptable

Author Response

Response to reviewer 3 Comments

Reviewer #3 
The manuscript is now acceptable

Response: We would like to thank reviewer 3 for their time and effort in reviewing this manuscript. We hope that our work will help contribute to drug development against Zika virus and other flaviviruses, benefiting the readers of Viruses journal.

Sincerely,
Ruben Abagyan
March 26, 2023
